# 2D Monomolecular Nanosheets Based on Thiacalixarene Derivatives: Synthesis, Solid State Self-Assembly and Crystal Polymorphism

**DOI:** 10.3390/nano10122505

**Published:** 2020-12-14

**Authors:** Alena A. Vavilova, Pavel L. Padnya, Timur A. Mukhametzyanov, Aleksey V. Buzyurov, Konstantin S. Usachev, Daut R. Islamov, Marat A. Ziganshin, Artur E. Boldyrev, Ivan I. Stoikov

**Affiliations:** 1A.M. Butlerov Chemical Institute, Kazan Federal University, Kremlevskaya Street, 18, 420008 Kazan, Russia; anelia_86@mail.ru (A.A.V.); padnya.ksu@gmail.com (P.L.P.); timur.mukhametzyanov@kpfu.ru (T.A.M.); abuzurov95@gmail.com (A.V.B.); marat.ziganshin@kpfu.ru (M.A.Z.); boldyrev25@gmail.com (A.E.B.); 2Institute of Fundamental Medicine and Biology, Kazan Federal University, Kremlevskaya Street, 18, 420008 Kazan, Russia; k.usachev@kpfu.ru; 3A.E. Arbuzov Institute of Organic and Physical Chemistry, Russian Academy of Sciences, Arbuzov Street, 8, 420088 Kazan, Russia; daut1989@mail.ru

**Keywords:** nanomaterials, 2D nanostructures, thiacalix[4]arene, terpenoids, geraniol, X-ray crystal analysis, 2D monomolecular-layer nanosheets, polymorphism

## Abstract

Synthetic organic 2D materials are attracting careful attention of researchers due to their excellent functionality in various applications, including storage batteries, catalysis, thermoelectricity, advanced electronics, superconductors, optoelectronics, etc. In this work, thiacalix[4]arene derivatives functionalized by geranyl fragments at the lower rim in *cone* and *1,3-alternate* conformations, that are capable of controlled self-assembly in a 2D nanostructures were synthesized. X-ray diffraction analysis showed the formation of 2D monomolecular-layer nanosheets from synthesized thiacalix[4]arenes, the distance between which depends on the stereoisomer used. It was established by DSC, FSC, and PXRD methods that the obtained macrocycles are capable of forming different crystalline polymorphs, moreover dimethyl sulphoxide (DMSO) is contributing to the formation of a more stable polymorph for *cone* stereoisomer. The obtained crystalline 2D materials based on synthesized thiacalix[4]arenes can find application in material science and medicine for the development of modern pharmaceuticals and new generation materials.

## 1. Introduction

In recent years, two-dimensional (2D) materials have attracted a lot of attention from researchers due to their superior functionality, such as mechanical, electrical, magnetic and optical properties [1,2,3,4]. In accordance to their structure, composition and electronic properties 2D materials may have to categorized as thin materials (graphene, silicene, germanene, and their saturated forms; hexagonal boron nitride; silicon carbide), rare earth, semimetals, transition metal chalcogenides and halides, and finally synthetic organic 2D materials, exemplified by 2D covalent organic frameworks [5]. Among the large family of these materials, synthetic organic 2D materials occupy a special place, which are used to create three-dimensional (3D) hybrid superlattices from alternating layers of organic molecules [1]. Creation of 3D superlattices provides new opportunities in various applications, including batteries, catalysis, thermoelectricity, advanced electronics, superconductors, optoelectronics, etc. According to the literature [1,5], the search for organic structures for creating 2D materials is still a difficult problem, since in many cases the orientation of organic molecules can be disordered. As a consequence, 2D materials based on them tend to repackage or aggregate, which leads to a weakening or even disappearance of 2D characteristics. Therefore, obtaining organic molecules capable of forming a monolayer or bilayer nanostructures with ordered packing is an urgent task for creating 3D superlattices with a fixed distance between layers, which plays a decisive role in the characteristics of the materials obtained.

In addition, recently there has been risen sharply the interest to polymorphic systems as an essentially interesting phenomenon and as an increasingly important component in the development and marketing of different materials based on organic molecules. The search of possible polymorphs is important for pharmaceuticals [6], high-energy materials, dyes and pigments [7] because different polymorphs usually have different properties [8]. Polymorphism significantly affects the physicochemical properties of materials, such as stability, density, melting point, solubility, bioavailability, etc. [9]. Hence, characterization of all possible polymorphs, identification of a stable polymorph and the development of a reliable process for sequential production are crucial significance in modern design of pharmaceuticals and materials.

In this work, we propose to use molecules with conformationally flexible and restricted units for creating 2D material. The controlled self-assembly of 2D structures is implemented using the example of well-known phospholipids, which are capable of forming monomolecular and bimolecular layers [10,11,12,13]. To obtain such structures, terpenoid fragments, which are found among phospholipids, are of particular interest. It is known that terpenoids such as verbenol [14] and menthol [15] are capable of polymorphism and crystalline modifications. Therefore, geraniol, an available non-toxic natural monoterpene, was chosen as such fragment. In addition, its structure is conformationally limited due to the presence of a double bond. As a result, compounds containing geraniol fragments are capable of packing and organizing into mono- and bilayers [16,17]. Amphiphilic compounds containing a polar fragment are usually used to obtain these layers, similar to surfactants, both ionic and non-ionic. Nonionic surfactants, along with anionic and cationic ones, were used to design 2D nanocrystals [18]. In this work, we propose to use a macrocyclic fragment of thiacalix[4]arene to create structures capable of forming mono- and bimolecular layers. The advantage of this platform is the presence of various stereoisomeric forms [19,20,21,22,23,24,25] allowing geranyl fragments to be strictly oriented in the space. Thiacalix[4]arenes are conformationally flexible molecular platforms that allow varying the size of the internal cavity, the number, nature and the spatial arrangement of binding sites. Calixarene derivatives are widely used for construction of different functional materials. Among such examples are known hybrid organic-inorganic silica nanoparticles for separation of model oligonucleotides and proteins [26], biodegradable, biocompatible polylactide polymers with cyclophanes as a core [27], antibacterial and catalytic systems based on calixarenes [28,29]. There are examples of thiacalixarene derivatives for the development of synthetic receptors for biologically important anions [22] and dopamine [23], which can be used in biomimetic materials, diagnostic agents and in the treatment of neurodegenerative diseases. Mention should also be made about polyplexes based on thiacalixarene derivatives, which are used for DNA packaging and have promising applications in biomolecular delivery systems and long-term storage of human genetic material [25]. Among other things, the controlled (smart) formation of polymorphs based on thiacalixarene derivative are also known [30]. Thus, according to previous contributions and applications of thiacalix[n]arenes it may be concluded that these macrocycles are promising candidates for constructing 2D nanostructures, especially different polymorphs that can be used in medicine for the development of modern pharmaceuticals, diagnostic agents and new generation materials.

This work is devoted to the synthesis of thiacalix[4]arene derivatives functionalized at the lower rim with geranyl fragments in *cone* and *1,3-alternate* conformations, the development of approaches to the formation of 2D monomolecular nanosheets based on them in a crystal, and the study of possible polymorphic transitions in the solid phase (Figure 1). 

## 2. Materials and Methods

### 2.1. General

Melting points were determined using the Boetius Block apparatus. All chemicals were purchased from Aldrich and most of them used as received without additional purification. Organic solvents were purified by standard procedures. The ^1^H and ^13^C NMR spectra were recorded on the Bruker Avance 400 spectrometer (400.0 MHz for H-atoms and 100.0 MHz for C-atoms), chemical shifts are reported in ppm (See Appendix A). The residual solvent peaks were used as an internal standard (CDCl_3_). The concentration of the analyzed solutions was 10 mM. The ATR FTIR spectra were recorded on the Spectrum 400 (Perkin Elmer, Seer Green, Lantrisant, UK) IR spectrometer: resolution 1 cm^−1^, accumulation of 64 scans, recording time 16 s; in the range of wave numbers 400–4000 cm^−1^ (See Appendix A). Electrospray ionization mass spectra (ESI) were obtained on an AmazonX mass spectrometer (Bruker Daltonik GmbH, Bremen, Germany) (See Appendix A). All experiments were conducted in negative ion polarity mode, in the range of *m*/*z* from 100 to 2800. The voltage on the capillary was −4500 V. Nitrogen was used as the gas-drier with a temperature of 300 °C and a flow rate of 10 L min^−1^, and nebulizer pressure of 55.16 kPa. Data was processed using DataAnalysis 4.0 (Bruker Daltonik GmbH, Bremen, Germany). The crystallinity of the samples were additionally controlled using BXFM polarizing optical microscope with LC30 camera (Olympus, Tokyo, Japan) (See Appendix A).

### 2.2. Synthesis

#### 2.2.1. Geranyl Bromoacetate (**2**)

A solution of 0.02 mol of geraniol and 0.021 mol of DIPEA in chloroform (120 mL) was prepared and cooled to −5 °C. A thermometer was installed to control the temperature. A solution of 0.021 mol of bromoacetic acid bromide in 10 mL of chloroform was added dropwise at such a rate that the temperature did not rise above −2 °C. After the addition, the reaction mixture was left for 1 h at room temperature. The resulting reaction mixture was then washed with 5% aqueous Na_2_CO_3_ solution (2 × 50 mL) and 50 mL of water. The organic solvent was removed on a rotary evaporator under reduced pressure. The resulting crude product was purified by column chromatography on silica gel (eluent hexane-propanol-2 at ratio 20:1). The yield of pale yellow oil of **2** was 4.29 g (78%). ^1^H NMR (CDCl_3_, δ, ppm, J/Hz): 1.60 (3H, s, CH_3_), 1.68 (3H, s, CH_3_), 1.71 (3H, s, CH_3_), 2.04–2.11 (4H, m, CH_2_–CH_2_), 3.83 (2H, s, O=C–CH_2_Br), 4.68 (2H, d, =CH–CH_2_–O, ^3^*J*_HH_ = 7.2 Hz), 5.07 (1H, m, =CH), 5.35 (1H, m, =CH). ^13^C NMR (CDCl_3_, δ, ppm): 16.54, 17.71, 25.70, 26.11, 26.22, 39.52, 63.13, 117.36, 123.61, 131.93, 143.58, 167.23. IR spectrum (ν/cm^–^^1^): 2967 (–CH_3_), 2915 (–CH_2_–), 2856 (–CH_3_), 1735 (C=O), 1444 (=CH–), 1424 (=CH–), 1408 (=CH–), 1376 (–CH_3_), 1276 (C–O–C), 1204 (C–O–C), 1153 (C–O–C), 953 (=CH–), 888 (=CH–), 830_._(=CH–), 554 (–CH_2_Br). Calcd for C_12_H_19_BrO_2_ (%): C, 52.38; H, 6.96; Br, 29.04. Found (%): C, 52.13; H, 6.77; Br, 28.90.

#### 2.2.2. 5,11,17,23-Tetra-*Tert*-Butyl-25,26,27,28-Tetra[(Geranyloxycarbonyl)-Methoxy]-2,8,14,20-Tetrathiacalix[4]arene (*Cone-***4**)

A mixture of 1.00 g (1.38 mmol) of 5,11,17,23-tetra-*tert*-butyl-25,26,27,28-tetrahydroxy-2,8,14,20-tetrathiacalix[4]arene (**3**), 3.06 g (11.11 mmol) of geranyl bromoacetate (**2**), 1.18 g (11.11 mmol) of anhydrous sodium carbonate in 70 mL of dry acetone was refluxed for 100 h. After cooling the precipitate from the reaction mixture was filtered off and washed with 2 × 10 mL acetone. The filtrate was concentrated under reduced pressure to give an oil that was recrystallized from methanol and cooled to −20 °C. The precipitate formed was filtered off and dried in vacuo over P_2_O_5_. The yield of thiacalix[4]arene *cone-***4** (white powder) was 0.83 g (40%). Mp: 53–56 °C. ^1^H NMR (CDCl_3_, δ, ppm, J/Hz): 1.08 (s, 36H, (CH_3_)_3_C); 1.59 (s, 12H, –CH_3_); 1.67 (s, 24H, –CH_3_); 2.04 (m, 8H, –CH_2_–CH_2_–); 2.08 (m, 8H, –CH_2_–CH_2_–); 4.65 (d, 8H, –O–CH_2_–CH=, ^3^*J*_HH_ = 7.0 Hz); 5.08 (br.t, 4H, =CH–CH_2_–CH_2_–, ^3^*J*_HH_ = 7.0 Hz); 5.16 (s, 8H, –O–CH_2_–); 5.34 (br.t, 4H, =CH–CH_2_–O–, ^3^J_HH_ = 7.0 Hz); 7.27 (s, 8H, Ar–H). ^13^C NMR (CDCl_3_, δ, ppm): 16.64, 25.85, 26.50, 31.29, 39.72, 61.70, 70.85, 118.44, 123.98, 129.50, 134.21. ^1^H–^1^H NOESY NMR spectrum (most important cross-peaks are presented): H^4b^/H^3^, H^9^/H^10^, H^9^/H^12^, H^10^/H^13^, H^12^/H^13^, H^13^/H^15^, H^14^/H^17^, H^15^/H^17^. IR spectrum (ν/cm^–^^1^): 2962 (–CH_3_), 2911 (–C_Ph_–H), 1760 (C=O), 1730 (–C(O)OR), 1669 (C=C), 1477 (–CH_2_–), 1443 (–CH_3_), 1432 (–CH_2_–CH=C–), 1382 (–CH_3_), 1266 (C=O), 1177 (–C(O)OR), 1098 (–C(O)OR), 1054 (C_Ph_–O–CH_2_–). MS (ESI): calculated [M^+^] *m/z* = 1496.8, found [M + Na]^+^
*m/z* = 1519.9. El. Anal. Calcd for C_88_H_120_O_12_S_4_ (%): C, 70.55; H, 8.07; S, 8.56. Found (%): C, 70.43; H, 7.94; S, 8.79. 

#### 2.2.3. 5,11,17,23-Tetra-*Tert*-Butyl-25,26,27,28-Tetra[(Geranyloxycarbonyl)-Methoxy]-2,8,14,20-Tetrathiacalix[4]arene (*1,3-Alternate-***5**)

A mixture of 0.50 g (0.69 mmol) of 5,11,17,23-tetra-*tert*-butyl-25,26,27,28-tetrahydroxy-2,8,14,20-tetrathiacalix[4]arene (**3**), 1.53 g (5.55 mmol) of geranyl bromoacetate (**2**), 1.76 g (5.55 mmol) of anhydrous cesium carbonate in 50 mL of dry acetone was refluxed for 30 h. After cooling, the precipitate from the reaction mixture was filtered off and washed with 2 × 10 mL acetone. The filtrate was concentrated under reduced pressure to give an oil that was recrystallized from methanol and cooled to −20 °C. The precipitate formed was filtered off and dried in vacuo over P_2_O_5_. The yield of thiacalix[4]arene *1,3-alternate-***5** (white powder) was 0.56 g (54%). Mp: 96–103 °C. ^1^H NMR (CDCl_3_, δ, ppm, J/Hz): 1.24 (s, 36H, (CH_3_)_3_C); 1.60 (s, 12H, -CH_3_); 1.69 (s, 12H, –CH_3_); 1.72 (s, 12 H, –CH_3_); 2.06 (m, 8H, –CH_2_–CH_2_–); 2.10 (m, 8H, –CH_2_–CH_2_–); 4.63 (s, 8H, –O–CH_2_–); 4.70 (d, 8H, –O–CH_2_–CH=, ^3^*J*_HH_ = 7.0 Hz); 5.09 (br.t, 4H, =CH–CH_2_–CH_2_–, ^3^*J*_HH_ = 6.4 Hz); 5.37 (br.t, 4H, =CH–CH_2_–O–, ^3^*J*_HH_ = 7.0 Hz); 7.51 (s, 8H, Ar–H). ^13^C NMR (CDCl_3_, δ, ppm): 16.69, 17.87; 25.86, 26.45, 31.22, 34.31; 39.75, 61.64, 68.44; 118.39, 123.90, 127.78, 132.05; 133.80; 146.31; 157.44; 168.04. ^1^H–^1^H NOESY NMR spectrum (most important cross-peaks are presented): H^3^/H^9′^, H^4b^/H^3^, H^4b^/H^7′^, H^4b^/H^9′^, H^4b^/H^10′^, H^4b^/H^12′^, H^9^/H^12^, H^10^/H^9^, H^10^/H^13^, H^10^/H^14^, H^12^/H^13^, H^14^/H^17^, H^15^/H^13^, H^15^/H^14^, H^15^/H^17^. IR spectrum (ν/cm^-1^): 2964 (–CH_3_), 2909 (–C_Ph_–H), 1764 (–C(O)OR), 1676 (C=C), 1479 (–CH_2_–), 1443 (–CH_3_), 1424 (–CH_2_–CH=C–), 1383 (–CH_3_), 1269 (C=O), 1184 (–C(O)OR), 1086 (–C(O)OR), 1062 (C_Ph_–O–CH_2_–). MS (ESI): calculated [M^+^] *m*/*z* = 1496.8, found [M + K]^+^
*m*/*z* = 1536.8. El. Anal. Calcd for C_88_H_120_O_12_S_4_ (%): C, 70.55; H, 8.07; S, 8.56. Found (%): C, 70.32; H, 8.05; S, 8.17.

### 2.3. Differential Scanning Calorimetry (DSC)

The DSC experiments were performed using the DSC204 F1 Phoenix differential scanning calorimeter (Netzsch, Selb, Germany) in an argon atmosphere (flow rate 150 mL/min) with the heating/cooling rate of 10 K/min. The cooling–heating cycle was repeated twice. DSC204 F1 Phoenix was calibrated according to the manufacturer’s recommendations by measuring six standard compounds (Hg, In, Sn, Bi, Zn, and CsCl) as described previously [31]. The error in the temperature and enthalpy determination by this technique were 0.1 K and 3%, respectively. The onset temperatures of processes were determined using the DTG curve. Samples of calixarenes with a mass of 1.73 mg (**5**) and 3.36 mg (**4**) were placed in a 40 μL aluminum crucible with a lid containing 0.5 mm diameter hole.

### 2.4. Fast Scanning Calorimetry (FSC)

FSC experiments were carried out on a FlashDSC 1 instrument (Mettler-Toledo, Greifensee, Switzerland) using one and the same UFS1 calorimetric chip-sensor for all measurements [32]. Prior to the experiments, the sensor was conditioned and corrected according to the manufacturer’s recommendations and calibrated using biphenyl and benzoic acid as standards. The measurements were performed at 27 °C sensor-support temperature and in argon dynamic atmosphere at 80 mL/min. The state of the samples was monitored by an optical microscope BXFM (Olympus, Tokyo, Japan), under crossed polarizers.

### 2.5. Powder X-ray Diffraction (PXRD) Experiment 

X-ray powder diffractograms were determined using a MiniFlex 600 diffractometer (Rigaku, Tokyo, Japan) equipped with a D/teX Ultra detector. In this experiment, CuK_α_ (*λ* = 1.54178 Å) radiation (30 kV, 10 mA) was used, and K_β_ radiation was eliminated with a Ni filter. The diffractograms were determined at RT in the reflection mode, with a scanning speed of 5°/min. The clathrate samples were loaded into a glass holder. The patterns were recorded without sample rotation.

### 2.6. X-ray Diffraction (XRD)

Data sets for single crystals **4** and **5** were collected on a Rigaku XtaLab Synergy S instrument (Rigaku, Tokyo, Japan) with a HyPix detector and a PhotonJet microfocus X-ray tube using Cu K_α_ (1.54184 Å) radiation at low temperature. Images were indexed and integrated using the CrysAlisPro data reduction package. Data were corrected for systematic errors and absorption using the ABSPACK module: numerical absorption correction based on Gaussian integration over a multifaceted crystal model and empirical absorption correction based on spherical harmonics according to the point group symmetry using equivalent reflections. The GSPF reference application line (GRAL) module was used for analysis of systematic absences and space group determination. The structures were solved by direct methods using SHELXT (University of Göttingen, Göttingen, Germany) [33] and refined by the full-matrix least-squares on F^2^ using SHELXL [34]. Non-hydrogen atoms were refined anisotropically. The hydrogen atoms were inserted at the calculated positions and refined as riding atoms. The positions of the hydrogen atoms of methyl groups were found using rotating group refinement with idealized tetrahedral angles. The figures were generated using Mercury 4.1 program. The *t*-Bu fragment in crystals **5** was located with the aid of rigid fragment [35] and subsequently refined with constraints and restraints.

Crystal Data for **4.** C_88_H_120_O_12_S_4_, M_r_ = 1498.07, triclinic, *P*-1 (No. 2), *a* = 13.9848(3) Å, *b* = 17.1054(4) Å, *c* = 19.9615(4) Å, *α* = 107.312(2)°, *β* = 105.052(2)°, *γ* = 98.286(2)°, *V* = 4273.55(18) Å^3^, *T* = 99.97(15) *K*, *Z* = 2, *Z’* = 1, *µ*(Cu K_α_) = 1.475, 62,647 reflections measured, 17386 unique (R_int_ = 0.0551) which were used in all calculations. The final wR_2_ was 0.3254 (all data) and R_1_ was 0.0999 (*I* > 2(*I*)). CCDC number: 2045930.

Crystal Data for **5.** C_88_H_120_O_12_S_4_, M_r_ = 1498.07, orthorhombic, *Ccce* (No. 68), *a* = 20.2109(11) Å, *b* = 32.012(2) Å, *c* = 13.3175(7) Å, *α* = *β* = *γ* = 90°, *V* = 8616.4(9) Å^3^, *T* = 100.00(10) K, *Z* = 4, *Z’* = 0.25, *µ*(Cu *K_α_*) = 1.463, 14,969 reflections measured, 4359 unique (*R_int_* = 0.0401) which were used in all calculations. The final *wR_2_* was 0.4798 (all data) and *R_1_* was 0.1451 (*I* > 2(*I*)). CCDC number: 2045929.

## 3. Results

### 3.1. Synthesis of Thiacalix[4]arene Derivatives

The synthesis of targeted compounds capable of self-organization in monolayers is an important and current challenge of modern organic chemistry, and also has practical application in materials science and nanotechnology. A suitable platform for creating such structures is *p*-*tert*-butylthiacalix[4]arene [36,37]. The functionalization of this macrocycle by terpenoid fragments allows one to obtain structures with a rigid framework and conformationally flexible substituents, potentially capable of self-organization.

Initially, an alkylating agent geranyl bromoacetate was obtained according to the original method. The desired product was obtained by the reaction of geraniol **1** with bromoacetic acid bromide in chloroform in the presence of diisopropylethylamine (DIPEA) (Scheme 1). Geranyl bromoacetate **2** was synthesized in good 78% yield and purified by column chromatography on silica gel using hexane/propanol-2 at 20:1 ratio as eluent. The structure of geranyl bromoacetate **2** was confirmed by a complex of physical methods, including ^1^H, ^13^C NMR, IR spectroscopy and elemental analysis.

In order to obtain tetrasubstituted thiacalix[4]arene derivatives containing terpenoid fragments, the interaction of *p*-*tert*-butylthiacalix[4]arene **3** with geranyl bromoacetate in acetone in the presence of alkali metal (sodium, cesium) carbonates was studied (Scheme 1). The base and solvent were specified in accordance with their efficiency in alkylation of *p-tert*-butylthiacalix[4]arene at the lower rim [38].

As expected, the targeted tetrasubstituted *p*-*tert*-butylthiacalix[4]arenes in *cone*
**4** and *1,3-alternate*
**5** conformations were obtained using sodium and cesium carbonates as base, respectively. Low yields (40 and 54%) of the compounds **4** and **5** are concerned with recrystallization and purification steps.

The structure and composition of the new thiacalix[4]arene derivatives **4** and **5** were characterized by ^1^H, ^13^C NMR, IR spectroscopy, mass spectrometry (ESI) and elemental analysis. The spatial structure of the synthesized macrocycles was established by 2D NOESY NMR spectroscopy.

In the ^1^H NMR spectra of the obtained compounds the signals of *tert*-butyl and aromatic protons appear as singlets at 1.08 ppm (macrocycle **4**), 1.24 ppm (macrocycle **5**) and 7.27 ppm (macrocycle **4**), 7.51 ppm (macrocycle **5**), respectively, which indicates the formation of symmetrical products (See Appendix A). The signals of oxymethylene protons are also observed as singlets in the area of strong fields (4.63 ppm) for *1,3-alternate*
**5** stereoisomer compared to *cone*
**4** stereoisomer, the signals of which are observed at weaker fields (5.16 ppm). Apparently, this is due to the shielding of oxymethylene protons by aromatic fragments of the macrocycle in *1,3-alternate* conformation. The signals of the methyl protons of the substituent are observed as singlets, the signals of the methylene protons of the substituent are observed as multiplets, and the signals of the methine protons at the double bond are observed as a broadened triplet. Chemical shifts, multiplicity, and integrated intensity of proton signals in the ^1^H NMR spectrum are in good agreement with the proposed structures of *p*-*tert*-butylthiacalix[4]arenes.

The spatial structure of the compounds **4** and **5** was studied by two-dimensional NMR ^1^H–^1^H NOESY spectroscopy (See Appendix A). The absence in the 2D NMR NOESY spectrum of the compound **4** of cross peaks due to the dipole-dipole interaction of the *tert*-butyl group protons with oxymethylene protons, as well as cross peaks between the aromatic protons of the macrocycle and oxymethylene protons, confirms that macrocycle **4** is in *cone* conformation (See Appendix A). The presence of cross peaks in the 2D NMR NOESY spectrum of compound **5** due to the dipole-dipole interaction of the protons of the *tert*-butyl group with methyl, oxymethylene and methine protons at the double bond, as well as cross peaks between the aromatic protons of the macrocycle and oxymethylene protons, confirms that the macrocycle **5** is in *1,3-alternate* conformation (See Appendix A).

In the IR spectra of the obtained compounds **4** and **5** (See Appendix A), an absorption band of stretching vibrations of the ether group (*ν*, 1750–1790 cm^–1^), the carbonyl group (*ν*, 1260–1269 cm^−1^) and double bonds (*ν*, 1669–1676 cm^−1^) were observed. They were absent in the IR spectrum of the initial thiacalix[4]arene **3**. In the IR spectra of thiacalix[4]arenes **4** and **5** the absorption bands of valence (*ν*, 2962–2964 cm^−1^) and deformation vibrations of the methyl group (*δ_as_*, 1443; *δ_s_*, 1382–1383 cm^−1^) were observed. The absence of the absorption band of stretching vibrations of the hydroxyl group in the IR spectra of compounds **4** and **5** indicates the complete substitution of the initial *p*-*tert*-butylthiacalix[4]arene **3**.

The spatial structure of the obtained products **4** and **5** was fully confirmed using X-ray (Figure 2 and Figure 3). In both cases, suitable single crystals were obtained from a DMSO solution. The symmetry group of product **4** is *P*-1 (Figure 2). As can be seen from Figure 2B, in the case of compound **4**, the formation of a zigzag close packing is observed, and in one layer the orientation of the molecules in one direction “head to tail”, and in the next layer the orientation is opposite to “tail to head”. The distance between these layers is 8 Å. Due to this, the layers are superimposed on each other in a zigzag manner, forming a close packing. The distance between the layers has a subnanometer size. 

The symmetry group of product **5** is *Ccce* (Figure 3). As can be seen from Figure 3C, in the case of compound **5**, the formation of strictly ordered nanostructured layers is observed, the distance between which is equal to half of the axis *b*, which is approximately 16 Å. The elementary cell is a rectangle. Based on these X-ray analysis, the structure of thiacalix[4]arene **5** in the *1,3-alternate* conformation is characterized by high-order symmetry. Moreover, the fragment shown in Figure 3B is independent, all the others are symmetrical to it.

### 3.2. DSC Analysis

Since macromolecular structures stabilized by the interaction of many weak forces undergo conformational or phase transitions upon heating, significant information about these structures can be obtained using DSC analysis. The thermal properties of thiacalix[4]arenes **4** and **5** obtained were studied by DSC, Figure 4. The results obtained indicate different thermal properties of the studied thiacalixarenes. Macrocycle **4** does not show any polymorphic transitions during the first heating up to melting point at 52.6 °C, Figure 4A. The enthalpy of melting is 27.3 J/g.

The DSC curve (first heating) obtained for the thiacalixarene **5** has a more complex shape. Above 91 °C, the first endothermic process begins, on which the second endothermic process is overlapped. The total enthalpy of the two processes is 39 J/g. Such behavior is characteristic of calixarenes capable of polymorphism [30,39], and may be due to the presence of two polymorphic forms of **5**. The studied calixarenes do not crystallize on cooling the melts. On reheating, thiacalixarene **4** retains a supercooled liquid state, Figure 4A, while thiacalixarene **5** undergoes several transformations, Figure 4B. The DSC curve at 56 °C exhibits an exothermic effect with Δ*H* = 21.9 J/g, which corresponds to the phase transition of calixarene from the amorphous state to the crystalline state (cold crystallization). The resulting crystalline phase melts at 95 °C with Δ*H* = 24.9 J/g. The temperatures of the peaks *T*_p_ and the end *T*_f_ of melting in the first and second heating runs are almost identical: *T*_1p_ = 105.9 °C, *T*_2p_ = 106.4 °C and *T*_1f_ = 109.1 °C, *T*_2f_ = 108.7 °C. Thus, last endothermic effect corresponds to the melting of the same polymorphic modification of thiacalixarene. The lower value of the enthalpy of melting on the second heating can be associated with the partial melting of calixarene at a lower temperature due to the heat of cold crystallization. The overlap of two opposite processes also explains the lower value of the enthalpy of cold crystallization in comparison with the enthalpy of melting during the first heating.

### 3.3. PXRD Data

The complex shape of the DSC curve of thiacalix[4]arene **5** indicates the presence of several polymorphic modifications of the macrocycle. To confirm the presence of polymorphs, the method of powder X-ray diffraction analysis was applied (Figure 5).

According to PXRD data (Figure 5A,B), macrocycle **4** does not undergo any conformational transformations after melting of the sample (at 60 °C). This is in good agreement with the DSC data, where no polymorphic forms of thiacalix[4]arene **4** were observed. In the case of thiacalix[4]arene **5** to control the temperature and phase state the sample was heated and cooled using DSC, then the same sample was reheated to 80 °C. The resulting samples were analyzed by PXRD (Figure 5D,E).

Diffraction patterns 5c and 5d differ from each other at an angle of 2*θ* = 20°, which confirms the presence of a polymorphic transition of macrocycle **5** according to DSC data. In this case, after cooling and reheating to 80 °C (Figure 5E), the diffraction pattern remains identical. Apparently, this is due to the formation of a stable polymorph of macrocycle **5** after the melting of the sample.

### 3.4. Fast Scanning Calorimetry

Compared to the conventional DSC, FSC provides faster temperature scanning rates both on cooling and heating and uses much smaller samples (typically 10–1000 ng). The latter trait is of particular interest in the investigation of the compounds with polymorphism [39].

The quality of the calorimetric curve in FSC greatly depends on the thermal contact between the sample and the chip. Typically, the studied compound will be “premelted,” i.e., melted and crystallized so that the resulting crystal has a good and uniform contact surface with the chip [40]. This approach may not be suitable for the compounds with polymorph transitions because the melting may destroy the polymorph of interest, and another, more stable of faster forming polymorph may be produced upon cooling. Taking the above considerations into account, the experiments were performed as follows.

In the first step of the investigation, the scans of the powders after synthesis were performed for both studied compounds (Figure 6A,B).

The endotherm of melting is visible in scans of both compounds. The irregular shape of the endotherm can be the result of the poor thermal contact between the sample and the chip, as well as the presence of several polymorph forms at the same time. No thermal effects were visible on the cooling scans and on the second heating scans in both cases.

Next, the samples were crystallized by adding a speckle of the crystals of the respective compound to the amorphous material produced in the previous step of research. The growth of the crystalline phase was monitored by the polarized optical microscopy (POM). The FSC scans of the developed crystals are shown in Figure 7A,B.

As seen on the graphs, the melting endotherms have a regular shape with the onset temperature, which is in agreement with the DSC results. However, in the case of compound **5**, a complex melting effect was seen in some scans (See Appendix A). This appears to be stochastic and can be a result of the presence of different polymorphs in the recrystallized sample. Thus, the added seeds may promote growth of different polymorphs simultaneously.

We have also performed FSC scans of the crystals grown from DMSO solutions. Prior to scanning, the solvent was evaporated at 45 °C, which is sufficient due to the high surface to volume ratio of the samples used for FSC. Full evaporation of the solvent was confirmed by monitoring the repeatability of the heat flow in **5** heating and cooling scans between 30 to 45 °C. The final FSC melting scans of the crystals grown from the DMSO solution are shown in Figure 8A,B.

The melting effect of compound **5** (Figure 8B) is in agreement with the result of the experiments with the recrystallized samples, while in the case of the compound 4, the melting effect is visible at a much higher temperature (Figure 8A). A small peak at lower temperature, which is close to the position of the peak of the recrystallized sample (Figure 7A), is also visible. Thus, compound 4 is also able to form different crystal polymorphs. The formation of the more stable polymorph is promoted by DMSO.

The geranyl substituents are located on the one side of the macrocyclic ring in *cone* stereoisomer **4** and on the opposite sides of the macrocycle in *1,3-alternate* stereoisomer **5**. Like that, in the case of *cone*-**4**, four geranyl substituents are spatially separated from *tert*-butyl fragments, and in the case of *1,3-alternate*-**5**, geranyl fragments alternate with bulky *tert*-butyl groups. As a result, the compounds **4** and **5** have different symmetry of the molecules and different crystalline packaging that provide the formation of different crystalline polymorphs. The above description is in good agreement with X-Ray data. In crystalline form of *cone*-**4** the orientation of the molecules of one layer in one direction “head to tail” alternates with the orientation of next layer “tail to head”. Contrary, the structure of *1,3-alternate*-**5** is characterized by high-order symmetry and the molecules overlap each other like “stacks” to form a channel. Since the structures of the synthesized compounds contain geranyl fragments capable of conformational rotations and, consequently, changes in their packing, it is obvious that the obtained thiacalixarenes are capable of the existence different polymorphic forms. It was shown by the results of FSC scans that both compounds can form different crystal polymorphs. The bulky nature of the substituents results in a slow crystallization process, which prevents monotropic polymorph transitions or melting-crystallization-melting effects as seen in faster crystallizing materials [41].

## 4. Conclusions

Thus, for the first time, an approach for the preparation of 2D nanostructures in crystal based on novel thiacalix[4]arene derivatives functionalized at the lower rim with geranyl fragments in *cone* and *1,3-alternate* conformations was developed. X-ray diffraction analysis showed the formation of 2D monomolecular-layer nanosheets from the thiacalix[4]arenes synthesized. It was found that the distance between the layers in the case of *1,3-alternate* stereoisomer has a nanometer size (16 Å), and in the case of *cone* stereoisomer subnanometer size (8 Å). It was established by DSC, PXRD and FSC methods that the obtained macrocycles are capable of forming different crystalline polymorphs. It was shown that DMSO promotes the formation of a more stable polymorph in the case of thiacalix[4]arene with geranyl fragments in *cone* conformation. However, in the case of *1,3-alternate* stereoisomer, a similar stable polymorph is formed in a single crystal (recrystallization from DMSO) and in crystals obtained by melt crystallization. Finding possible polymorphs is important for pharmaceuticals, high-energy materials, dyes and pigments. Therefore, the obtained stable polymorphs of thiacalix[4]arenes synthesized, which form an ordered packing of monomolecular 2D nanosheets, can find application in nanomaterials science and medicine for the modern development of pharmaceuticals and new generation materials.

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
