# Peer review of "2D Monomolecular Nanosheets Based on Thiacalixarene Derivatives: Synthesis, Solid State Self-Assembly and Crystal Polymorphism"

_nanomaterials, 2020, doi:10.3390/nano10122505_

Round 1
Reviewer 1 Report
This contribution is a comprehensive study of a calix[4]arene derivative self-organized into 2D layers. The synthesis and characterization is well-described and very clear. The interest of the system described for the materials chemistry and nanoscience community is remarkable.
My recommendation is to publish this paper in Nanomaterials. I only have a few suggestions:
A thorough revision of previous contributions and applications of calix[n]arenes in the revision could help to put the contribution in a more specific perspective.
The authors describe in a generic manner many potential applications of these systems (e.g. nanomaterials science and medicine for the modern development of pharmaceuticals and new generation materials...). Whilst this may be true is too general, some more specific topics could be covered, which may be link to the above mentioned revision of the litterature of calix[n]arenes.
Finally, some attention is requited to a few items. For instance, change ml by mL, insert an space between numbers and º (e.g. change 56ºC by 56 ºC).
Author Response
This contribution is a comprehensive study of a calix[4]arene derivative self-organized into 2D layers. The synthesis and characterization is well-described and very clear. The interest of the system described for the materials chemistry and nanoscience community is remarkable.
My recommendation is to publish this paper in Nanomaterials. I only have a few suggestions:
- A thorough revision of previous contributions and applications of calix[n]arenes in the revision could help to put the contribution in a more specific perspective.
The authors describe in a generic manner many potential applications of these systems (e.g. nanomaterials science and medicine for the modern development of pharmaceuticals and new generation materials...). Whilst this may be true is too general, some more specific topics could be covered, which may be link to the above mentioned revision of the literature of calix[n]arenes.
Answer:
Thank you for comment. In the Introduction the following text was added:
“Calixarene derivatives are widely used for construction of different functional materials. Among such examples are known hybrid organic-inorganic silica nanoparticles for separation of model oligonucleotides and proteins [26], biodegradable, biocompatible polylactide polymers with cyclophanes as a core [27], antibacterial and catalytic systems based on calixarenes [28,29]. There are examples of thiacalixarene derivatives for the development of synthetic receptors for biologically important anions [22] and dopamine [23], which can be used in biomimetic materials, diagnostic agents and in the treatment of neurodegenerative diseases. Mention should also be made about polyplexes based on thiacalixarene derivatives, which are used for DNA packaging and have promising applications in biomolecular delivery systems and long-term storage of human genetic material [25]. Among other things, the controlled (smart) formation of polymorphs based on thiacalixarene derivative are also known [30]. Thus, according to previous contributions and applications of thiacalix[n]arenes it may be concluded that these macrocycles are promising candidates for constructing 2D nanostructures, especially different polymorphs that can be used in medicine for the development of modern pharmaceuticals, diagnostic agents and new generation materials.”
The corresponding references also were added:
- Yuskova, E. A., Ignacio-de Leon, P. A. A., Khabibullin, A., Stoikov, I. I., Zharov, I. Silica nanoparticles surface-modified with thiacalixarenes selectively adsorb oligonucleotides and proteins. J. Nanopart. Res. 2013, 15(10), 2012. https://doi.org/10.1007/s11051-013-2012-8
- Mostovaya, O. A., Gorbachuk, V. V., Padnya, P. L., Vavilova, A. A., Evtugyn, G. A., Stoikov, I. I. Modification of Oligo- and Polylactides With Macrocyclic Fragments: Synthesis and Properties. Front. Chem. 2019, 7, 554. https://doi.org/10.3389/fchem.2019.00554
- Rodik, R. V., Boyko, V. I., Kalchenko, V. I. Calixarenes in bio-medical researches. Curr. Med. Chem. 2009, 16(13), 1630-1655. https://doi.org/10.2174/092986709788186219
- Schühle, D. T., Peters, J. A., Schatz, J. Metal binding calixarenes with potential biomimetic and biomedical applications. Coord. Chem. Rev. 2011, 255(23-24), 2727-2745. https://doi.org/10.1016/j.ccr.2011.04.005
- Finally, some attention is requited to a few items. For instance, change ml by mL, insert an space between numbers and º (e.g. change 56ºC by 56 ºC).
Answer:
The corrections mentioned by reviewer were done in the text.

Reviewer 2 Report
This manuscript contains the synthesis of thiacalix[4]arene derivatives functionalized by geranyl fragments at the lower rim in cone and 1,3-alternate conformations, which are capable of controlled self-assembly in a 2D nano-structures. The background as well as the results are well presented and the authors take great care to limit solid state self-assembly and crystal polymorphism. The obtained crystalline 2D materials based on synthesized thiacalix[4]arenes can be applied in material science and medicine for the development of modern pharmaceuticals and new generation materials. Thus, this manuscript could be interesting for the readers in the field of supramolecular chemistry as well as material sciences. Also this paper could well serve as the basis for a development of these fields.
Minor comments
- The obtained thiacalix[4]arenes with geranyl fragments formed different crystalline polymorphs depending on the cone- and 1,3-alternate-conformation. The authors should pay some comments to explain these findings.
- In Sheme 1 the authors should improve the drawing the reagents and conditions (up and down the arrow).
- In the experimental section the authors should describe the appearances of the compounds synthesized.
I recommend this manuscript for publication in Nanomaterials after minor revision described above.
Author Response
This manuscript contains the synthesis of thiacalix[4]arene derivatives functionalized by geranyl fragments at the lower rim in cone and 1,3-alternate conformations, which are capable of controlled self-assembly in a 2D nano-structures. The background as well as the results are well presented and the authors take great care to limit solid state self-assembly and crystal polymorphism. The obtained crystalline 2D materials based on synthesized thiacalix[4]arenes can be applied in material science and medicine for the development of modern pharmaceuticals and new generation materials. Thus, this manuscript could be interesting for the readers in the field of supramolecular chemistry as well as material sciences. Also this paper could well serve as the basis for a development of these fields.
Minor comments
- The obtained thiacalix[4]arenes with geranyl fragments formed different crystalline polymorphs depending on the cone- and 1,3-alternate-conformation. The authors should pay some comments to explain these findings.
Answer:
The explanation about different crystalline polymorphs depending on the conformation of synthesized compounds was added in the text:
“The geranyl substituents are located on the one side of the macrocyclic ring in cone stereoisomer 4 and on the opposite sides of the macrocycle in 1,3-alternate stereoisomer 5. Like that, in the case of cone-4, four geranyl substituents are spatially separated from tert-butyl fragments, and in the case of 1,3-alternate-5, geranyl fragments alternate with bulky tert-butyl groups. As a result, the compounds 4 and 5 have different symmetry of the molecules and different crystalline packaging that provide the formation of different crystalline polymorphs. The above description is in good agreement with X-Ray data. In crystalline form of cone-4 the orientation of the molecules of one layer in one direction “head to tail” alternates with the orientation of next layer “tail to head”. Contrary, the structure of 1,3-alternate-5 is characterized by high-order symmetry and the molecules overlap each other like “stacks” to form a channel. Since the structures of the synthesized compounds contain geranyl fragments capable of conformational rotations and, consequently, changes in their packing, it is obvious that the obtained thiacalixarenes are capable of the existence different polymorphic forms. It was shown by the results of FSC scans that both compounds can form different crystal polymorphs.”
- In Scheme 1 the authors should improve the drawing the reagents and conditions (up and down the arrow).
Answer:
Thank you for comment. The Scheme 1 was improved in accordance with the reviewer comments.
- In the experimental section the authors should describe the appearances of the compounds synthesized.
Answer:
Thank you for comment. The following corrections eremade in the experimental section:
“The yield of pale yellow oil of 2 was 4.29 g (78%).”
“The yield of thiacalix[4]arene cone-4 (white powder) was 0.83 g (40%).”
“The yield of thiacalix[4]arene 1,3-alternate-5 (white powder) was 0.56 g (54%).”
